# Peripartum Traumas and Mental Health Outcomes in a Low-Income Sample of NICU Mothers: A Call for Family-Centered, Trauma-Informed Care

**DOI:** 10.3390/children10091477

**Published:** 2023-08-30

**Authors:** Allison Williams, Anna Beth Parlier-Ahmad, Erin Thompson, Rachel Wallace, Paul B. Perrin, Alyssa Ward, Karen D. Hendricks-Muñoz

**Affiliations:** 1Department of Psychiatry, Virginia Commonwealth University Medical Center, Richmond, VA 23298, USA; 2Department of Psychology, Virginia Commonwealth University, Richmond, VA 23284, USA; parlierab@vcu.edu; 3Center for Children and Families, Florida International University, Miami, FL 33199, USA; erthomps@fiu.edu; 4Department of Psychology, Hunter Holmes McGuire VA Medical Center, Richmond, VA 23249, USA; rachel.wallace@va.gov; 5Department of Psychology, University of Virginia, Charlottesville, VA 22903, USA; perrin@virginia.edu; 6Department of Neonatology, Virginia Commonwealth University Medical Center, Richmond, VA 23298, USA; dralyssaward@gmail.com (A.W.); karen.hendricks-munoz@vcuhealth.org (K.D.H.-M.)

**Keywords:** postpartum mental health, NICU infant, post-traumatic stress, adverse childhood experiences

## Abstract

Postpartum depression (PPD), postpartum anxiety (PPA), and post-traumatic stress disorder (PTSD) among birthing people have increased substantially, contributing to adverse maternal/infant dyad outcomes, with a high prevalence in the neonatal intensive care unit (NICU). Despite calls for trauma-informed care in the NICU and high rates of post-traumatic stress, little research has examined the rates of or the relationships between peripartum mood and adverse child experiences (ACEs) in NICU mothers or evaluated which peripartum traumas are most distressing. This study employed structural equation modeling (SEM) to explore whether peripartum-related traumas and NICU-related stressors mediated the associations between ACEs and mental health outcomes in 119 lower-income, racially diverse mothers in a Level IV NICU. Mental health concerns were prevalent and highly comorbid, including 51.3% PPA, 34.5% PPD, 39.5% post-traumatic stress, and 37% with ≥4 ACEs. The majority (53.8%) of mothers endorsed multiple peripartum traumas; NICU admission was the most common trauma (61%), followed by birth (19%), pregnancy (9%), and a medical event in the NICU (9%). Our SEMs had good fit and demonstrated that ACEs predicted peripartum distress. Trauma-informed care efforts should employ transdiagnostic approaches and recognize that women commonly present to the NICU with childhood trauma history and cumulative peripartum traumas.

## 1. Introduction

The prevalence of perinatal mood and anxiety disorders (PMADs), as well as post-traumatic stress, has increased substantially over the past decade, contributing to adverse maternal/infant dyad outcomes [1]. Maternal distress is heightened when infants are hospitalized, which can constitute a traumatic event for mothers. Moreover, the inherent stress of a neonatal intensive care unit (NICU) admission contributes to higher rates of peripartum mental health concerns among mothers with infants in the NICU, including post-traumatic stress, postpartum depression (PPD), and postpartum anxiety (PPA) [2,3]. A nuanced understanding of perinatal trauma and mental health sequelae is needed to provide trauma-informed, family-centered care [4].

Compared to other medical populations, NICU mothers may be at heightened risk for more severe mental health symptoms, given the potential exposure to multiple peripartum traumas along the peripartum reproductive arc (pre-pregnancy, pregnancy, labor and delivery, and postpartum) prior to their infant’s hospitalization [5,6]. For example, some medical problems that lead to NICU admission are diagnosed prenatally (e.g., genetic conditions and cardiac diagnoses), which can contribute to fears about the infant’s development and well-being throughout the pregnancy. High-risk pregnancies are associated with postpartum and post-traumatic stress, depression, and anxiety, and mental health concerns during pregnancy can be compounded by a potentially traumatic birth [7]. NICU mothers are also more likely to perceive childbirth as traumatic compared to well-baby mothers. In addition to potentially traumatic pregnancies and births, the NICU itself can be traumatic, including both NICU admission as well as medical events while on the unit [8].

In NICU mothers, post-traumatic stress is common, with rates of early post-traumatic stress (known as Acute Stress Disorder, ASD) ranging from 5–30%, compared to 1–2% of well-baby mothers [9,10,11]. Additionally, rates of both PPD (8–40%) and PPA (18–43%) are considerably higher and more persistent in NICU mothers compared to well-baby mothers [12,13,14]. Furthermore, post-traumatic stress in NICU mothers is often comorbid with other mental health concerns, including PPD and PPA [12]. However, few studies have examined the relations among PPD, PPA, and post-traumatic stress in NICU mothers.

Experiencing high ACEs (six or more) is associated with nine times the risk for NICU admission [15,16]. Moreover, according to a systematic review, exposure to ACEs has been associated with perinatal anxiety and depression, even after controlling for sociodemographic, psychiatric, and psychosocial risk factors [17]. Despite the strong evidence for prior cumulative stress predicting preterm birth, the literature is scarce on ways in which prior chronic stress exposure can influence post-traumatic stress after NICU hospitalization. To date, no studies have evaluated the relationship between ACEs and mental health outcomes in NICU mothers specifically.

Despite the potential for cumulative trauma exposure, research on NICU post-traumatic stress anchors post-traumatic stress questionnaires with “NICU admission” as the index trauma, rather than allowing for parents to identify alternative potentially traumatic aspects of their experience (e.g., the birth and a code in the unit). The singular focus on NICU hospitalization does not account for the endorsement of multiple perinatal traumas outside of the NICU experience. The cumulative experience of ACEs, a potentially traumatic pregnancy, and a traumatic childbirth experience put NICU mothers in a vulnerable position as they experience the hospitalization of their infant on top of these prior experiences [2]. Moreover, NICU clinicians themselves may be less likely to recognize the NICU as traumatic, necessitating research that bridges the gap between parent experiences and provider perceptions [18]. Given these risk factors, additional research is needed to examine how various potential traumas prior to and along the peripartum reproductive arc are perceived and may relate to postpartum mental health outcomes.

To date, research has not evaluated ways in which perceptions of trauma influence post-traumatic stress and other mental health sequalae. More specifically, there is a dearth of literature on trauma in NICU mothers, particularly during the first month of hospitalization, despite the potential mental health consequences such as ASD, PPD, and PPA. In order to improve family-centered, trauma-informed maternity care, research must examine how experiencing traumatic events in rapid succession may worsen mental health outcomes and how exposure to prior childhood traumas (i.e., ACEs) may influence the peripartum- and NICU-associated mental health concerns.

The primary aims of the current study were to (1) uncover the prevalence rates of ACEs, peripartum trauma perceptions, and mental health issues in NICU mothers during the first month of hospitalization, with a focus on index trauma (i.e., peripartum trauma identified as the worst), and (2) explore whether peripartum-related traumas (i.e., pregnancy, birth, and transition to NICU) and NICU-related stressors (i.e., Sights and Sounds, Role Alteration, and Infant Appearance) mediated the associations between ACEs and mental health outcomes among NICU mothers. Findings from the current study have the potential to inform theories of stress sensitization as well as provide future direction for trauma-informed care.

## 2. Materials and Methods

### 2.1. Participants and Design

This was a medical record chart review and cross-sectional survey study on a 40-bed Level IV NICU with private infant rooms, located in an academic medical center. **Inclusion criteria included** English-speaking biological mothers with an infant hospitalized in the NICU who were eligible for participation between days 5 and 31 of their infant’s hospitalization. **Exclusion criteria** included death during the first month of NICU admission, admission less than 5 days, and non-English-speaking mothers.

Data collection took place in the unit primarily from July 2018 to January 2019. Clinical psychology doctoral research assistants approached potential participants in their infant’s private NICU room, obtained consent, and administered the survey battery verbally, as detailed below. The survey battery took approximately 30 min to 1 h. To reduce participant burden, demographic and infant health variables were extracted from the infant chart. Participants were offered the opportunity for additional psychological follow-up visits throughout their stay in the unit, following the completion of their intake survey. This study was conducted as a portion of a doctoral candidate’s dissertation; additional procedural details and a publication focused specifically on infant health are available online [19]. This study received IRB approval.

### 2.2. Measures

All scales are publicly accessible and do not require permission for use unless otherwise noted.

**Adverse Childhood Experiences (ACEs)** were assessed using the 10-item *ACEs Questionnaire*, a measure of childhood trauma, with summed scores ranging from zero to ten (scores of four or greater are considered “high risk”) [20].

**Acute Stress Disorder (ASD)** was examined using the 22-item *Impact of Events Scale-Revised (IES-R)*, which assesses post-traumatic stress over a 7-day period, with summed scores ranging from 0 to 88 and subscales of Intrusion, Avoidance, and Hypervigilance [21]. Scores of 24 to 32 represent “at risk”, where ASD is a potential clinical concern, and scores of 33 and above suggest “probable ASD diagnosis”. To avoid obtaining symptoms related to a non-perinatal trauma, we used only perinatal trauma (pregnancy, birth, NICU admission, and NICU events) as the index trauma.

**Postpartum Anxiety** over a two-week period was assessed with the *Generalized Anxiety Disorder 2-item* (GAD-2) [22]. A cutoff score of greater than or equal to three has good sensitivity (86%) and specificity (83%) for Generalized Anxiety Disorder (GAD).

**Postpartum Depression** over a two-week period was assessed using the *Patient Health Questionnaire-2* (PHQ-2) item version [23]. A cutoff score of greater than or equal to three has good sensitivity (83%) and specificity (90%) for Major Depressive Disorder (MDD).

**NICU Stressors** were assessed using *the NICU Parental Stressor Scale (PSS)*, a 46-item scale with 3 subscales measuring perceived stress related to various domains of NICU hospitalization, with higher scores representing greater distress [24]. Domains included “Sights and Sounds” (e.g., noises of monitoring equipment/alarms), “Role Alteration” (e.g., not being allowed to hold or care for the baby), and “Infant Appearance and Behavior” (e.g., the baby’s fragile appearance/size). Written permission for use was obtained by scale developers.

**Peripartum Traumatic Events** (index and compounding) were specified by having mothers indicate the single most distressing aspect of their infant’s progression to the NICU (index trauma) and identify any other traumatic components (compounding traumas). Options for index and other peripartum traumas included pregnancy, birth, transition to NICU (first few days of hospitalization), or a NICU health incident. Endorsed index trauma items were used to anchor responses to the IES-R.

### 2.3. Data Analyses

Descriptive statistics were generated using SPSS 24. Between <1% and 10% of variables had missing data. Missing data were addressed using the expectation–maximization algorithm in SPSS 24 and were confirmed to be missing at random using Little’s missing completely at random (MCAR) test (IBM Corp., Armonk, NY, USA, 2016).

### 2.4. Primary Analyses

Descriptive statistics were conducted to examine the prevalence of and correlations among indices of trauma exposure (i.e., childhood trauma, peripartum traumas, and NICU-related stressors) and mental health outcomes (i.e., postpartum anxiety, depression, and Acute Stress Disorder).

A mediational SEM was conducted to examine the direct effects of ACEs (ACE total score), cumulative NICU traumas, and NICU distress on mental health outcomes (PPA, PPD, and ASD). Indirect effects among model variables were also examined.

The SEM was conducted with a 2000 bootstrap sample in AMOS software [25]. To examine indirect effects, we calculated *p*-values via bootstrap approximation (conducted via two-sided, bias-corrected confidence intervals). We assessed goodness of fit using comparative fit index (CFI) values >0.95, goodness of fit (GFI) values of >0.90, adjusted goodness of fit (AGFI) indices of >0.90, a root mean square error of approximation (RMSEA) value of <0.08, and a standardized root mean square residual (SRMR) value of <0.10 [26,27].

## 3. Results

### 3.1. Descriptive Statistics

Of the eligible mothers with infants in the NICU (*n* = 124), 96% (*n* = 119) provided informed consent. Participants were 48.7% Black, 39.5% White, 7.6% Hispanic/Latinx, and 2.5% Asian. Educational attainment included 5% who did not complete high school, 7.6% with a general education diploma, 37.8% with a high school degree, 15.1% with some college coursework, 21% with a college degree, and 12.6% with a master’s degree or higher. The majority of infants (71.8%) had Medicaid insurance. Infant gestational age included the following: 13.4% extremely preterm (<28 weeks), 16% very preterm (28–32 weeks), 47.1% moderate to late preterm (32–37 weeks), and 20% term (>37 weeks). Additional infant health information is included in a publication from this sample focused specifically on infant health [19].

### 3.2. Aim One: Prevalence and Association of Exposure to Trauma, NICU Distress, and Risk of Adverse Maternal Mental Health

Descriptive statistics for exposure to trauma (i.e., ACEs and peripartum trauma), overall NICU distress, and maternal mental health (i.e., ASD, PPD, and PPA) are presented in Table 1.

Correlations between the maternal mental health and trauma exposure constructs are presented in Table 2. Overall, participants’ cumulative ACEs and peripartum trauma scores were significantly correlated with ASD, PPA, and PPD. When examining associations between the various types of peripartum trauma exposure and mental health outcomes, only birth trauma was significantly correlated with ASD, PPA, and PPD.

### 3.3. Aim 2: Associations between Trauma Exposure and Maternal Mental Health Outcomes

Direct and indirect effects were examined among ACEs, peripartum trauma, NICU-related stressors, ASD, PPD, and PPA. Figure 1 shows a visual representation of the model, along with direct effects and their statistical significance. The *χ*^2^ value was not statistically significant, suggesting a good model fit, *χ*^2^ = (11, *n* = 119) = 19.082, *p* = 0.06. Other fit indices also indicated adequate to good fit: CFI = 0.982, GFI = 0.964, AGFI = 0.881, and RMSEA = 0.079 (CI, 0.000, 0.137). Overall, this model explained 59.5% of the variance in ASD symptoms, 36.4% in PPD symptoms, and 35.6% in PPA symptoms.

The mothers’ cumulative ACE scores directly predicted ASD (*B* = 0.27, *p* < 0.05), PPD (*B* = 0.27, *p* < 0.05), and PPA (*B* = 0.27, *p* < 0.05). ACE scores also significantly predicted cumulative peripartum trauma scores (*B* = 0.20, *p* < 0.05) and NICU-related stressors (*B* = 0.23, *p* < 0.05).

ACE scores had a significant indirect effect on NICU-related distress via cumulative peripartum trauma score (*B* = 0.055, *p* = 0.018). Additionally, NICU-related stressors mediated the associations between peripartum trauma score and ASD (*B* = 0.167, *p* < 0.001), PPD (*B* = 0.058, *p* = 0.006), and PPA (*B* = 0.092, *p* = 0.001). Finally, cumulative peripartum trauma scores and NICU-related distress mediated the associations between the mothers’ cumulative ACE scores and ASD (*B* = 0.161, *p* = 0.017), PPD (*B* = 0.118, *p* = 0.006), and PPA (*B* = 0.120, *p* = 0.004).

## 4. Discussion

Despite the strong evidence for prior cumulative stress predicting preterm birth, there is a dearth of literature on the associations between trauma exposure and mental health risk in NICU mothers. To address this gap in the literature, the current study examined the prevalence of and associations between childhood and peripartum trauma exposure and multiple mental health outcomes (i.e., ASD, PPD, and PPA) among mothers whose infants had recently been admitted to the NICU. The high prevalence of ASD symptoms, multiple traumas, and other mental health comorbidities found in the current study highlight the unique psychosocial needs of NICU mothers and underscore the importance of recognizing the interplay between childhood trauma, multiple peripartum traumas, and mental health outcomes.

### 4.1. Peripartum Mental Health in the NICU

Peripartum mood disorders and ASD were highly prevalent and comorbid in this sample. Approximately half of the mothers (*n* = 119) reported clinically significant symptoms of PPA (51.3%), and over a third of the mothers reported clinically significant PPD (34.5%) and ASD (39.5%). These rates are comparable to previous NICU mother studies, and as anticipated, much higher than well-baby rates [10]. Of mothers meeting the criteria for ASD, ~81% endorsed PPA and ~66% endorsed PPD. The high co-occurrence of these disorders underscores the global distress of mothers with peripartum mental health concerns and highlights the opportunity and need for transdiagnostic treatment approaches [28].

In this sample, mothers experienced moderate distress or higher across all NICU-related domains, with Role Alteration as the domain with the highest distress level, followed by Infant Behavior and Procedures, and Sights and Sounds [24]. Moreover, distress in all three domains was correlated with ASD, PPA, and PPD. These findings support the prior literature establishing a strong relationship between NICU-related distress and mental health sequelae [12,28]. Role alteration was also most highly correlated with peripartum mood disorders and post-traumatic stress. The assessment of specific NICU stressors may inform targeted interventions (e.g., enhance maternal role to prevent PPD). Additional research is needed to examine the impact of family-centered care practices on specific distress domains and related mental health sequelae.

### 4.2. Adverse Childhood Experiences among NICU Mothers

This is one of the first studies to examine the prevalence of ACEs in NICU parents. Many mothers (37%) experienced four or more ACEs, which is generally considered the threshold for poorer mental and physical health outcomes [20]. From a clinical perspective, understanding that NICU mothers have a high prevalence of ACEs underscores the importance of inclusion of trauma-informed care within the current standards of family-centered care in the NICU, as prior trauma is associated with an increased likelihood of post-traumatic stress responses following a subsequent traumatic event. Indeed, a previous systematic review revealed strong associations between ACEs and perinatal mental health concerns (specifically anxiety and depression), when controlling for other sociodemographic, psychiatric, and psychosocial risk factors [17]. Additionally, early motherhood is a period where mothers can reflect on their own childhoods and represents an opportunity to disrupt intergenerational trauma cycles and parenting practices.

### 4.3. Peripartum Traumas

Our findings provide a novel understanding related to the potential peripartum trauma parents perceive as the “most” traumatic peripartum event (i.e., index trauma). While NICU admission was the most commonly cited concern (61%), many (39%) participants reported other components of the peripartum experience, including labor and delivery (19%), high-risk pregnancy (9%), and a medical event on the NICU (9%). Results from the current study underscore the importance of directly assessing which aspect of the peripartum period is most traumatic to NICU mothers as an anchor for their symptoms, rather than presuming NICU admission is the index trauma.

Additionally, over half of the mothers (54%) endorsed multiple peripartum traumas. When looking at any trauma endorsed (either index or secondary trauma), 81% endorsed NICU hospitalization, 50% labor and delivery, 24% pregnancy, and 15% a specific NICU scare. The high rates of multiple peripartum traumas highlight the multifactorial trauma exposure and continuous trauma that NICU mothers experience. The experience of multiple peripartum traumas might contribute to the overall higher rates of post-traumatic stress in NICU parents compared to other medical populations, such as parents in the pediatric intensive care or oncology units [5].

### 4.4. Associations between Trauma Exposure and Mental Health Outcomes

We examined whether peripartum trauma and NICU-related stressors mediate the associations between ACEs and maternal mental health outcomes. Both ACEs and cumulative peripartum trauma exposure predicted mental health outcomes, suggesting that childhood and peripartum trauma exposure are important risk factors in predicting perceptions of the NICU and in understanding mental health outcomes. ACEs and peripartum trauma were the most strongly associated with ASD symptoms, compared to PPD and PPA. This is expected, as we would anticipate that increasing trauma exposure or endorsement of traumatic events would lead to greater post-traumatic stress.

### 4.5. Study Limitations

There are several significant study limitations inherent to study design, sample, and instrumentation. First, the data were cross-sectional. Although cross-sectional data are not always well-positioned to examine mediation, our constructs were still temporally sound, as ACEs preceded peripartum trauma exposure, which preceded NICU-related stressors. Furthermore, discretion should be exercised when inferring causality in SEM, especially when using cross-sectional data [29,30]. However, researchers have also shown that mediational and causal relationships in longitudinal SEMs can be similarly biased. As a result, readers should view the current SEMs and respective mediation effects with an appropriate degree of caution, given that no variables were or ethically could be manipulated. Study results would have been more generalizable with broader sample inclusion criteria, such as allowing non-English speaking families and including other caregivers. Additionally, this study sample may have higher psychosocial needs, given that it took place in a Level IV NICU serving a federally designated mental health shortage area and had a high portion of low-income mothers (~72% Medicaid). Additionally, while symptoms reflect mood concerns during the peripartum period, it is possible that many mothers in this sample additionally had mood concerns prior to hospitalization.

### 4.6. Trauma-Informed Care Implications and Conclusions

This study provides important future clinical implications for family-centered, trauma-informed clinical care. First, this study underscores the high prevalence of prior childhood trauma, cumulative peripartum traumas, and negative mental health outcomes among NICU mothers. These findings map onto recommended initiatives for trauma-informed care in the unit, specifically recognizing that families in the NICU might come from high trauma exposure backgrounds and that acute peripartum traumas should be considered cumulative [4]. To avoid re-traumatization, family-centered and trauma-informed care efforts must enhance maternal autonomy, proactively address mental health needs, facilitate bonding, and create opportunities for coregulation through enhancing and respecting maternal caregiving and de-medicalization of the infant. Additionally, asking parents to select which aspect of the peripartum arc was most traumatic for them—and allowing selection of multiple traumas if desired—prompts recognition in healthcare providers that there are many facets of the peripartum course that can impact the family and guide treatment targets. Moreover, the high correlations between NICU stressors and all three mental health concerns assessed underscore the importance of comprehensive assessment. For example, high scores in one domain (e.g., PPA) indicate the necessity of a full peripartum health evaluation. Finally, these symptoms were present within the first month on the unit, underscoring the importance of early identification, intervention, and treatment, ideally embedded within the unit. In this sample, only <5% had access to a mental health provider in the unit.

Given both the dearth of research on post-traumatic stress and PMADs in NICU mothers and the considerable consequences of negative maternal mental health for families, infants, and medical providers, it is critical to understand factors contributing to early maternal mental health concerns during the first month of admission. Findings from the present study underscore the prevalence of NICU stress, PMADs, and post-traumatic stress. Additionally, this study provides the first evaluation of cumulative peripartum trauma in NICU mothers, finding that a majority of NICU mothers experience multiple traumas in quick succession, which directly influence peripartum mental health concerns. The present study offers a model through which providers can better understand the development of post-traumatic stress, emphasizing the relations between trauma history, objective factors related to infant health and birth, and parental appraisals of these events. In particular, this is the first study to evaluate the influence of ACEs on post-traumatic stress in NICU parents, despite calls for ACE screening and trauma-informed care across medical settings [4]. Additionally, our findings note the critical importance of identifying how mothers perceive potential peripartum traumas to understand their post-traumatic stress responses. In sum, these findings underscore the high psychosocial needs of NICU mothers, as well as the importance of trauma-informed screening and intervention.

## Figures and Tables

**Figure 1 children-10-01477-f001:**
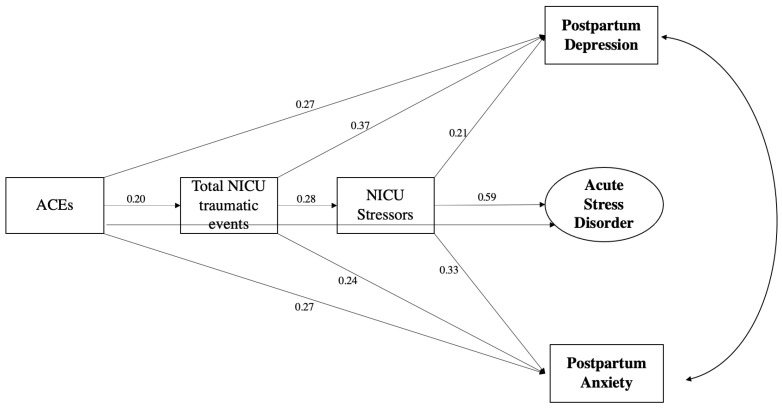
SEM Model of ACEs, peripartum trauma, and mental health outcomes.

**Table 1 children-10-01477-t001:** Trauma exposure and mental health in NICU mothers (*n* = 119).

Trauma Exposure and Mental Health Variable	Percent (Frequency)	Means ± SD
**Adverse Childhood Experiences**		
0	18.5% (*n* = 22)	
1	19.3% (*n* = 23)	
2	14.3% (*n* = 17)	
3	5% (*n* = 6)	
4 or more	37% (*n* = 39)	
**Peripartum trauma (primary or secondary)**		
Pregnancy	24.4% (*n* = 29)	
Birth	50.4% (*n* = 60)	
NICU hospitalization	80.7% (*n* = 96)	
Specific NICU scare	15.1% (*n* = 18)	
**Index (primary) peripartum trauma**		
Pregnancy	9.2% (*n* = 11)	
Birth	18.5% (*n* = 22)	
NICU hospitalization	61.3% (*n* = 73)	
Medical scare in the NICU	9.2% (*n* = 11)	
**Endorsement of multiple peripartum traumas**	53.8% (*n* = 64)	
**Acute Stress Disorder**		27.08 ± 20.02
Intrusion Subscale		12.27 ± 8.66
Hypervigilance Subscale		6.65 ± 5.86
Avoidance Subscale		8.16 ± 7.85
**Clinical severity**		
Not at risk for ASD	45.4% (*n* = 54)	
At risk for ASD	15.1% (*n* = 18)	
Probable ASD	39.5% (*n* = 47)	
**Anxiety**		3.00 ± 2.19
Subclinical or no anxiety	48.7% (*n* = 58)	
Anxiety	51.3% (*n* = 61)	
**Depression**		1.86 ± 2.08
Subclinical or no depression	64.7% (*n* = 77)	
Depression	34.5% (*n* = 41)	
**Comorbidities**		
ASD and depression	66.0% (*n* = 31)	
ASD and anxiety	80.9% (*n* = 38)	
**Overall NICU Distress**		
Sights and Sounds		9.82 ± 4.18
Infant Behavior and Procedures		34.56 ± 14.13
Role Alteration		23.58 ± 8.27

Note. *n* = 119. SD = standard deviation. NICU = neonatal intensive care unit. ASD = Acute Stress Disorder.

**Table 2 children-10-01477-t002:** Peripartum trauma and mental health correlations.

	ASD	Anxiety	Depression
Adverse Childhood Experiences	0.431	0.382 **	0.395 **
Cumulative peripartum traumas	0.295 **	0.304 **	0.325 **
Trauma during pregnancy	0.068	0.150	0.117
Birth trauma	0.223 *	0.188 *	0.245 **
Trauma during transition to the NICU	0.129	0.082	0.150
Trauma due to specific medical event in NICU	0.143	0.166	0.099

Note. * = *p* < 0.05; ** = *p* < 0.01. ASD = Acute Stress Disorder.

## Data Availability

Data for this study are not publicly available.

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
