# Peer review of "Peripartum Traumas and Mental Health Outcomes in a Low-Income Sample of NICU Mothers: A Call for Family-Centered, Trauma-Informed Care"

_children, 2023, doi:10.3390/children10091477_

Round 1

Reviewer 1 Report

Dear Authors

Very interesting paper but you have to pay attention to some points. The introduction section is a bit confusing. I suggest you start with your main topic, why NICU is a source of developing mental disorders for the mother? Then analyze how a mother experiences stress, what are the symptoms of depression and post-traumatic stress, and then refer to the bibliography.

The literature supporting the introduction is minimal. Need more sources from recent articles.

What was the number of participants? 

How did you rule out a pre-existing mental disorder? With which scale or with which questions were the revival of past trauma ruled out?

You also don't mention how you got permission to use these scales if you had to adapt them before...

Generally, you describe little of the methodology process which is a very basic part of an article.

In table 1, write the number of the sample

Studies of this kind cannot be cross-sectional. Require sample retesting and confounding variables. It is not easy to draw conclusions. Be more careful in your conclusions.

 Minor editing of English language required

Author Response

Very interesting paper but you have to pay attention to some points. The introduction section is a bit confusing. I suggest you start with your main topic, why NICU is a source of developing mental disorders for the mother? Then analyze how a mother experiences stress, what are the symptoms of depression and post-traumatic stress, and then refer to the bibliography.

Thank you for your suggestion- we have re-ordered the intro according to your feedback and feel it improved clarity. We have incorporated additional recent citations as well, including literature reviews and meta-analyses to provide richer background per your suggestion. 

What was the number of participants? 

119 participants, as described in results (96% consent rate). We’ve added the N=119 throughout the paper more throughout for ease in reading, as you noted this did get buried!

How did you rule out a pre-existing mental disorder? With which scale or with which questions were the revival of past trauma ruled out?

This is an important point of clarification-thanks for drawing it to our attention. We added clarifying language that the posttraumatic stress symptoms must be linked to a perinatal index trauma in the measures section. We also added a statement in our limitations section that prior mental health concerns could not be ruled out.

You also don't mention how you got permission to use these scales if you had to adapt them before...

We added a note detailing that all scales are publicly available with the exception of PSS, which received written permission by author developers.  

Generally, you describe little of the methodology process which is a very basic part of an article.

Thank you for your note- we have added some additional information around the methods, which were cross sectional survey while on the unit using the measures we outlined. We also added information and references regarding statistical analyses to make the methods more clear.

In table 1, write the number of the sample

Thank you for your comment, we have added the total sample number.

Studies of this kind cannot be cross-sectional. Require sample retesting and confounding variables. It is not easy to draw conclusions. Be more careful in your conclusions.

We thank the reviewer for pointing out the limitations of applying SEM to cross-sectional data. Although there is a wide variety of perspectives on this topic (West, 2011), Reichardt (2011) has shown that mediational inference in longitudinal SEMs can similarly be biased and even more directly, incorrect. As a result, we have now significantly minimized causal references throughout our manuscript when interpreting the results, included new references, and added language to our Limitations section.

Reviewer 2 Report

This hospital Record review study reflects on mental health of mothers whose babies were being care for in the NICU (Level IV). The subject of research is of interest to both patients and health providers as NICU admission of newborn have adverse outcomes in the immediate and long term. 

The Introduction reflects on a well searched literature review and justifies the need for such a study.  

Study design. This study is based on hospital records. The study design has its flaws but these have been addressed under 'limitations'. However, the authors have convinced the reader that mental health of mothers and affected based on both peripartum events and care in the NICU.

Although the authors have stated the inclusion criteria and have used research assistants to administer the various instruments to answer the research question, this section could be improved. 
We would suggest the authors explicitly state the research question and include inclusion and exclusion criteria for ease of reading. It would be useful to know if the questionnaires used in this study require permission for use . 

Kindly provide some references for the SEM modelling equation. 

Results: The tables and figures are well displayed. It would help the reader if details of perinatal trauma, reasons for admission to NICU, outcomes of the newborns after stay, morbidity and mortality statistics are included to appreciate the high prevalence of ACE and other mental issues. What is meant by perinatal trauma? Preterm birth first appears in the discussion. Are we referring to severe preterm births or termination of pregnancies for maternal safety e.g., severe preeclampsia with FGR etc. Were there any intrapartum traumatic experiences e.g., trauma due to forceps or cesarean deliveries?. 

Discussion. The authors have summarised the discussion points well. They have concluded based on the results. 
Limitations of the study and ethical approval has been stated.

All references are written well. 

Author Response

This hospital Record review study reflects on mental health of mothers whose babies were being care for in the NICU (Level IV). The subject of research is of interest to both patients and health providers as NICU admission of newborn have adverse outcomes in the immediate and long term. 

The Introduction reflects on a well searched literature review and justifies the need for such a study.  

Thank you!

Study design. This study is based on hospital records. The study design has its flaws but these have been addressed under 'limitations'. However, the authors have convinced the reader that mental health of mothers and affected based on both peripartum events and care in the NICU.

Although the authors have stated the inclusion criteria and have used research assistants to administer the various instruments to answer the research question, this section could be improved. 
We would suggest the authors explicitly state the research question and include inclusion and exclusion criteria for ease of reading. It would be useful to know if the questionnaires used in this study require permission for use . 

Thank you for the suggestion- we have added explicit inclusion/exclusion criteria. Additionally, we added statements on scale permissions. Finally, we fleshed out the methodology to be more clear.

Kindly provide some references for the SEM modelling equation. 

We have added 4 references accordingly.

Results: The tables and figures are well displayed. It would help the reader if details of perinatal trauma, reasons for admission to NICU, outcomes of the newborns after stay, morbidity and mortality statistics are included to appreciate the high prevalence of ACE and other mental issues. What is meant by perinatal trauma? Preterm birth first appears in the discussion. Are we referring to severe preterm births or termination of pregnancies for maternal safety e.g., severe preeclampsia with FGR etc. Were there any intrapartum traumatic experiences e.g., trauma due to forceps or cesarean deliveries?. 

This is a great question! We have addressed this by adding more details throughout the paper. First, we have a separate paper focused specifically on infant health outcomes and perinatal mental health, which includes a great deal of detail on infant health. To balance the reader’s need for additional sample information with wanting to avoid duplicating results, we have now linked this article for additional sample information. we highlighted here but are not going to present to avoid duplication We included additional language around perinatal trauma definitions as well throughout the paper.

Discussion. The authors have summarised the discussion points well. They have concluded based on the results. 
Limitations of the study and ethical approval has been stated.

All references are written well. 

Thanks for your thoughtful feedback!

Round 2

Reviewer 1 Report

Dear Authors 

You did a great job. Congratulations! 

Minor editing of English language required